# Effects of MWCNTs on Char Layer Structure and Physicochemical Reaction in Ethylene Propylene Diene Monomer Insulators

**DOI:** 10.3390/polym14153016

**Published:** 2022-07-26

**Authors:** Zhiheng Chen, Shida Han, Yuan Ji, Hong Wu, Shaoyun Guo, Ning Yan, Hongyan Li

**Affiliations:** 1The State Key Laboratory of Polymer Materials Engineering, Polymer Research Institute of Sichuan University, Chengdu 610065, China; chenzh104@163.com (Z.C.); sdhan1996@163.com (S.H.); jiyuan_scu@163.com (Y.J.); nic7702@scu.edu.cn (S.G.); 2Xi’an Modern Chemistry Research Institute, Xi’an 710065, China; chipsme@126.com

**Keywords:** polymer-matrix composites (PMCs), aramid fibers, MWCNTs, ablative resistance, thermal properties

## Abstract

As one of the most promising ablative fillers, multi-walled carbon nanotubes (MWCNTs) have been used to improve the ablative resistance of Ethylene–Propylene–Diene Monomer (EPDM) insulators by facilitating the carbothermal reduction reaction of silica. However, the contribution of MWCNTs to char layer structure of the insulators was unclear. In this work, the effects of MWCNTs on char layer structure and ablative resistance were investigated in different EPDM-based insulators with and without silica. The results showed that adding only 3 phr MWCNTs can reduce the linear ablation rate of EPDM-based insulators without silica by 31.7%, while 6 phr MWCNTs are required to obtain similar results in EPDM-based insulators with silica. The char layer morphology of the two insulators gradually evolved into a dense porous structure as MWCNTs content increased, but their formation mechanisms were different. The XRD and Raman spectrum showed that different physicochemical reactions occurred around MWCNTs under different charring components. The proposed ablation mechanism was further verified by designing alternating multilayer distribution of MWCNTs and silica. This work can guide the construction of desirable char layer structure for increasing the ablative resistance of EPDM-based insulators.

## 1. Introduction

Polymer-matrix composites (PMCs) have been commonly applied to the thermal protective system of Solid Rocket Motor (SRM) [1,2,3]. Ethylene–Propylene–Diene Monomer (EPDM) composites are considered the ideal insulators in combustion chamber of SRM due to their excellent ablative resistance, moderate thermal shock resistance, low density, and so on [4,5,6,7]. Generally, a porous char layer formed during the ablation can be the first barrier to resisting erosion in EPDM-based insulators [8,9,10,11]. Meanwhile, the erosion of heat flow and the escape of pyrolysis gas can be affected by the size and connection of the pores in char layers [12,13,14,15], and various ablative fillers (heat resistant fibers, ceramic fillers, phenolic resins and so on) are added to regulate the char layer structure for desirable ablative resistance [16,17,18]. However, with the advancement of propulsion technology, EPDM-based insulators need better ablative resistance to endure more severe conditions.

Multi-walled carbon nanotubes (MWCNTs) [19,20], graphene [21] and carbon nanoparticles [22,23,24] have performed excellently in several fields (like, thermal conductivity, electrical conductivity, catalysis, reinforcement, electromagnetic shielding, etc.), and thus, they were considered as highly effective ablative fillers. Especially MWCNTs with high strength and thermal conductivity are considered a promising ablative filler to improve the ablative resistance of EPDM-based insulators [25,26,27,28,29]. Li et al. [25,26] found that the addition of MWCNTs in the EPDM-based insulators accelerates the carbothermal reduction reaction of silica to increase the char residue, resulting in the improvement of the ablative resistance. In subsequent work, they [28] found that the char residues formed during the ablation can cover the surface of MWCNTs, which can effectively reduce the thermal conductivity of the char layers. In fact, the above ablation mechanisms were proposed based on the fact that the introduction of MWCNTs can affect the physicochemical reactions of silica during the ablation; however, the contribution of MWCNTs to char layer structure of EPDM-based insulators was unclear.

For phenolic composites, MWCNTs are regularly used to enhance the char layers to obtain excellent ablative resistance [30,31,32,33,34,35]. Wang et al. [32] emphasized the importance of MWCNTs networks on ablative resistance of phenolic composites and found that adding only 0.3% MWCNTs can decrease the linear ablation rate from 0.55 mm/s to 0.13 mm/s. Yum et al. [34] found that 0.1 wt% MWCNTs can reduce the linear ablation rate of phenolic composites by about 30% since the crystalline rods formed by the stress graphitization of MWCNTs can strengthen char layers. Additionally, Yazdani et al. [35] proposed that the char layers enhanced by the MWCNTs network can act as a barrier to protect the char residue, based on the result that adding 1.0 wt% MWCNTs can decrease the linear ablation rate of carbon fiber/phenolic composites by 80%. Inspired by the direct contribution of MWCNTs to the char layers of phenolic composites, the effect mechanism of MWCNTs on the char layers of EPDM-based insulators, especially without silica, should be crucial to fully utilize the ablative potential of MWCNTs.

In this work, different contents of MWCNTs were introduced to EPDM-based insulators with and without silica. The effects of MWCNTs on char layer structure and the physicochemical reactions during the ablation were investigated systematically under the two charring components. Furthermore, the ablation mechanism was proposed based on the relationship between the char layer structure and ablative resistance of EPDM-based insulators and further verified by designing alternating multilayer distribution of MWCNTs and silica in the insulators.

## 2. Experimental

### 2.1. Materials

A low Mooney viscosity EPDM (Keltan 2650C; ethylene content, 46.0 wt%; ENB content, 6.0%; ${ML}_{125{}^{\circ}C}^{1+4}$, 25) was purchased from LANXESS Germany. Chopped (6 mm) Kevlar fibers were supplied by Bluestar (Chengdu) New Material Co., Ltd. (Chengdu, China). The type of the MWCNTs (average diameter, 9.5 nm; average length, 1.5 μm) is the NC7000 purchased from Belgium. Fumed silica (average particle size, 5 μm) was supplied by Shijiazhuang Ruituo Chemical Technology Co., Ltd. (Shijiazhuang, China). Boron phenolic resins were obtained from Tianyu High-Temperature Resin Materials Co., Ltd (Bengbu, China). Bis(1-(tert-butylperoxy)-1-methylethyl)-benzene (BIPB) and triallylisocyanurate (TAIC) were supported by Hunan Yixiang Technology Co., Ltd. (Changsha, China) and Guangzhou Jinchangsheng Technology Co., Ltd. (Guangzhou, China), respectively. Paraffin oil, Sulphur, zinc oxide and stearic acid were supplied by Chengdu Chron Chemical Co., Ltd. (Chengdu, China). 

### 2.2. Preparation of EPDM-Based Insulators

Appendix A lists the formulations of EPDM-based insulators. Group NS and group S represent insulators without silica and with silica, respectively, and these insulators are abbreviated as NS-C-x and S-C-x, where x represents the content of MWCNTs. The additional content of MWCNTs was selected as 0 phr (parts per hundred grams of EPDM), 0.3 phr, 0.7 phr, 1.5 phr, 3 phr and 6 phr, which was set to facilitate the investigation of the effect of MWCNTs with low content on the ablative resistance of EPDM-based insulators. The preparation process mainly includes mixing and vulcanization. Firstly, EPDM, silica, MWCNTs, boron phenolic resin, Aramid Fiber (AF) and other fillers were mixed in a two-roll mix mill (LRM-S-150, Labtech, Beijing, China) at the roll temperature of 30 °C. When all ingredients were added, lower the roll gaps to less than 0.1 mm to obtain a good distribution. Subsequently, after standing for more than 24 h, the compounds can be vulcanized in the mold at 170 °C and 15 MPa for the optimum cure time T90.

### 2.3. Characterization and Tests

#### 2.3.1. Ablation Tests

Oxyacetylene ablation test was conducted on a test bed manufactured by Xi’an Zhi Rui Industrial System Engineering Co., Ltd. The flow rates of oxygen-acetylene gas in the test process were 1512 L/h and 1116 L/h, respectively. According to the calculation of the calibrated water-cooled heat flux meter, the actual value of heat flux reached 457 W/cm^2^. The test sample size and the distance from the sample surface to the tip of the nozzle are ф30 × 10 mm and 10 mm, respectively. The sample center was ablated with an oxyacetylene flame for 30 s. The mass ablation rate, linear ablation rate and charring rate are often used to evaluate the ablative resistance of insulators, and they are defined as follows. Mass ablation rate = (original mass of sample–mass of sample after ablation)/ablation time. Linear ablation rate = (original thickness of sample–thickness of sample after ablation)/ablation time. Charring rate = (original thickness of sample–thickness of sample without char layers after ablation)/working time.

In addition, a K-type thermocouple was set on the back face of the ablated sample to measure the maximum back-face temperature (T_max, b_) of insulators during the ablation process, which can be used to characterize the thermal insulation performance of the EPDM-based insulators.

#### 2.3.2. Thermal Stability and Thermal Conductivity Test

Thermogravimetric analysis (TGA) of MWNCTs and cured insulators were carried out from 30 °C to 800 °C with the heating rate of 10 °C/min under a nitrogen atmosphere, using a thermogravimetric analyzer (TG209F1 Iris. NETZSCH, Bavaria, Germany). Thermal conductivity of insulators was measured at room temperature using a transient plane heat source method apparatus (HotDisc1500, Sweden).

#### 2.3.3. Morphology Observation

The dispersion of MWCNTs in EPDM-based insulators was observed by transmission electron microscopy (TEM, Tecnai G2 F20 S-TWIN, FEI, State of Oregon, America). The char layer surface and char layer section were observed in detail by scanning electron microscopy (SEM, JSM-5900LV, Japan).

#### 2.3.4. Mechanical Property

An electronic universal testing machine (Model: CMT-4104, Shenzhen, China) was used to test the mechanical properties of the char layers and EPDM-based insulators.

#### 2.3.5. Component Analysis

Energy-dispersive X-ray spectrometer (EDS, Tokyo, Japan) was used to analyze the element content of char layers. The degree of crystallinity of the char layers with different content MWCNTs was analyzed by an X-ray diffractometer (XRD, Ultima IV, Rigaku, Tokyo, Japan) and Raman spectrum (LabRAM HR, HORIBA, Paris, France). 

## 3. Results and Discussion

### 3.1. Dispersion of MWCNTs and Thermal Stability of EPDM-Based Insulators

Appendix A show the dispersion of 3 phr MWCNTs in EPDM-based insulators with and without silica, respectively. Most MWCNTs have uniform dispersion in both EPDM-based insulators, except a few aggregates of MWCNTs, and it can be seen that silica formed larger clusters (Appendix A), showing poorer dispersion due to the high concentration used. As shown in Appendix A, it can be noticed that the TGA curve of MWCNTs and silica hardly changed at the temperature ranging from 30 °C to 800 °C, showing their excellent thermal stability, and the residual mass of both insulators gradually increases with the increase of MWCNTs content (Appendix A), suggesting the positive effects of MWCNTs on residual mass of the insulators, and the insulators of group S have higher residual mass than the insulators of group NS, which can be ascribed to the addition of silica with a high amount (Appendix A).

### 3.2. Ablative Resistance and Thermal Insulation Performance of EPDM-Based Insulators

The ablative resistance of the insulators can be directly represented by the linear ablation rate. As shown in Figure 1a and Appendix A, the linear ablation rate of both insulators showed different variation trends in two groups as MWCNTs content increased. In group NS, the linear ablation rate of the insulators decreased to 0.0801 mm/s as the MWCNTs content increased to 3 phr, which is 31.7% lower than that of the insulators without MWCNTs. However, the linear ablation rate of NS-C-6 is higher than that of the NS-C-3, indicating that the addition of excess MWCNTs is not conducive to the ablative resistance of the insulators in group NS. In group S, the linear ablation rate of the insulators increased from 0.0980 mm/s (S-C-0) to 0.1076 mm/s (S-C-0.7) as the MWCNTs content increased, which revealed that the addition of MWCNTs with low content can slightly decrease the ablative resistance of the insulators. When the MWCNTs content is over 0.7 phr, the linear ablation rate of the insulators begins to decrease as the MWCNTs content increases, and the S-C-6 shows the lowest linear ablation rate of 0.0837 mm/s in group S. Meanwhile, it can be noticed that the linear ablation rate of NS-C-3 is 9% lower than that of S-C-3 under the addition of MWCNTs with the same content, and even is 4.3% lower than that of S-C-6 which has the best ablative resistance in group S. The reason that MWCNTs have different effects on the ablative resistance in the EPDM-based insulators with and without silica are discussed by analyzing the char layer structure in the next section.

Due to their high thermal conductivity, the addition of MWCNTs is bound to have an impact on the heat transfer during the ablation. In this work, the thermal conductivity and the back-face temperature (T_max, b_) were used respectively to assess the heat transfer rates and thermal insulation performance. As can be seen from Figure 1b and Appendix A, the thermal conductivity and the T_max, b_ of the insulators in the two groups significantly increase with the increase in the MWCNTs content. The insulators of group S have higher thermal conductivity and T_max, b_, showing poorer thermal insulation performance than the insulators of group NS due to the addition of silica with high thermal conductivity of 27 W/mK. Interestingly, the NS-C-3 with the best ablative resistance has a lower T_max, b_ than that of all the insulators in group S, especially the T_max, b_ of NS-C-3 is even 3 °C lower than the S-C-0 which has the lowest T_max, b_ in group S. Additionally, the higher thermal conductance ability can accelerate the heat transfer from the surface to the interior and enhance the pyrolysis process of EPDM to promote the fast formation of char layers [12,15], and it can be noticed that the thickness of char layers increases with the increase in MWCNTs content in the insulators (Table 1).

### 3.3. Morphological Analysis and Component Analysis of Char Layers

The char layers are the first barrier of EPDM-based insulators to resist heat flow, and the char layer structure is closely related to the ablative resistance of EPDM-based insulators. Thus, analyzing the variation of char layer structure is essential to an in-depth understanding of the contribution of MWCNTs to ablative resistance in EPDM-based insulators. As shown in Figure 2, the char layer surface of group NS varied from a loose structure to a porous structure with the increase in MWCNTs content. It can be observed that the loose structure of NS-C-0 is formed based on the fiber skeleton arising from the exposed AF in Figure 2a,e. The loose structure can be easily invaded by the external heat flow, leading to the worst ablative resistance of NS-C-0. With the addition of MWCNTs, the high reactivity of MWCNTs promoted the deposition of pyrolysis gas [27,28,29,32,33], and some char residues appeared around the AF skeleton (Figure 2b,f) to improve the compactness of NS-C-0.3 char layer, and with the increase in MWCNTs, the accumulation of char residues based on AF skeleton is improved, making the char layer structure transform from looseness to porosity (Figure 2c,g), which enhances the ability to block heat flow and improves the ablative resistance. Furthermore, the char residues fixed by MWCNTs can prevent the carbon skeleton from contacting oxygen to protect it from oxidation [30,31,32,33], resulting in the improvement of oxidation resistance of the char layers. However, it can be found that the porous structure in char layers of NS-C-6 is denser than that of NS-C-3, while the NS-C-6 insulator shows a higher linear ablation rate. In fact, the char layers of NS-C-6 have smaller pore diameter than the char layers of NS-C-3 (Appendix A), and the char layers with smaller pore have higher pore pressure and need to be subject to more drastic internal erosion [8,10,11].

Since silica can melt and form liquid ceramic film to resist thermal–chemical erosion and promote the deposition of pyrolysis gas during ablation [8,18], porous char layers were formed based on carbonized matrix in group S, with interspersed AF to enhance the strength and integrity of char layers (as shown in Figure 3). For S-C-0, the char layers show a dense structure with a few pores (Figure 3a), while massive pores were generated on the char layers with the addition of a small amount of MWCNTs (such as Figure 3b and Appendix A), which shows a high degree of fragmentation. These char layers are conducive to the penetration of external heat into the interior, so the ablative resistance became worse compared to the S-C-0. Interestingly, the pore size of char layers became smaller with the gradual increase in MWCNTs (Appendix A), and the linear ablation rate started to decrease continually when the MWCNTs contents exceed 0.7 phr, which can be attributed to the effective blocking by progressively densified char layers. Meanwhile, elemental analysis was employed to evaluate the components on char layer surface, and it can be seen from Appendix A that the proportion of Si obviously increased due to the addition of MWCNTs, and the char layers with more MWCNTs have more proportion of Si (Appendix A), which revealed that the more gaseous silica was transformed to silicon carbide by the carbothermal reduction reaction under high-temperature condition [25,26], with less gaseous silica escaping from char layers.

The char layer cross-section of Appendix A are observed in magnification, and the image results are shown in Appendix A. The char layers of group NS are obviously loose and have many large pores (red circles in Appendix A), and the char layers of group S have only a few small pores and narrow cracks (red circles in Appendix A). The differences in the morphological structure of the char layers are still attributed to the variations of MWCNTs and silica. For group NS, similar to the evolution trend of their char layer surface, the pores of the char layer cross-section gradually decrease with the increase in MWCNTs content, and the reason can also be attributed to the fixation of pyrolytic carbon by MWCNTs [27,28,29,31,32,33]. Silica can promote the deposition of pyrolysis carbon by melting to form a liquid film [8,18], resulting in the dense char layers of group S. However, due to the coverage of much pyrolysis carbon, the char layer cross-section of group S is difficult to observe their evolutionary way.

Further magnification and observation of the char layers revealed the difference between the char layers of group NS and group S. The microstructure of char layers of group NS is shown in Appendix A, char residue increases to form a dense structure in char layers as MWCNTs content increasing, which is similar to the macroscopic evolution of the char layers in Figure 2. Likewise, the char layers of group S show finer and denser microstructure than that of group NS, and there are many smaller and uniform pores (Appendix A). The reason is that the carbothermal reduction reaction of silica can consume large chars to generate silicon carbide during the ablation [25,26]. It can be seen from Figure 4 that the dense structure in char layers is composed of micro networks, but the morphology and EDS results of the micro networks are different in both char layers. The micro networks in char layers of NS-C-3 are denser than that of S-C-3, and there are some aggregates of C element in the micro networks of char layers of NS-C-3 (Figure 4c). The Si and C element are uniformly distributed in the micro networks of S-C-3 (Figure 4f), which is because the most deposited chars are consumed by reacting with silica. Thus, the loose micro networks are formed inside the char layers of group S than that of group NS, which can explain the different effects of MWCNTs on ablative resistance in both EPDM-based insulators. Moreover, with the addition of MWCNTs with high content, many dense char residues appeared around the micro networks in the char layers of the two groups (Appendix A), which revealed the excellent ability of the micro networks to fix char residue. 

### 3.4. Compression Property and Component Analysis of Char Layers

Previous studies [25,26,27,29,30,31] proposed that the MWCNTs can form rod-shaped structure by promoting the graphitization of carbonized matrix to strengthen char layers, resulting in the increasing compression property of char layers. The results of the compression test (Table 1) of char layers in our work showed similar findings. As the MWCNTs content increased, char layers showed growing compression stress in two groups, and the char layers of group S have higher compression strength than that of group NS under the addition of MWCNTs with the same content since the MWCNTs can promote the formation of silicon carbide with high hardness [25,26]. 

XRD test (Figure 5) and Raman spectrum (Figure 6) were used to analyze the composition of char layer and the trend of I_D_/I_G_ (the ratio of intensities of D band around 1340 cm^−1^/G band near 1570 cm^−1^, which can on behalf of the metric of “an amount of crystal boundary” or the disorder of graphitic structure [31,34]) were further calculated for Raman spectra (shown in Figure 6c). It can be noticed that the (002) peak corresponding to near-graphite structures improved obviously as the increase in MWCNTs content (Figure 5a,b) [36,37], which indicated the ordered structure of carbon increased in the two charring components. As a matter of fact, MWCNTs were considered can act as crystal growth nuclei for the graphitization of carbonized matrix [31,34] and improve the content of near-graphite structure of char layers. However, the above effects are obviously different in the two charring components (Figure 6), and it can be observed that the I_D_/I_G_ of group NS rapidly decreased as the increase in MWCNTs, while the value of group S only slightly decreased. The results suggest that MWCNTs only can effectively promote the graphitization of pyrolysis carbon in the charring component without silica, while the MWCNTs are more trending to promote the carbothermic reduction of silica in the charring component with silica [31,32], and the formation of silicon carbide can also be proved by the XRD profiles of group S (Figure 5d), which showed the peak ((111), (220), (311)) corresponding to silicon carbide [26,31,34]. 

### 3.5. Ablation Mechanism

The difference in ablative resistance in EPDM-based insulators can be ascribed to the diversity of their char layer structure. Therefore, we attempt to explain the contribution of MWCNTs on ablative resistance in EPDM-based insulators by analyzing the effects of MWCNTs on the char layer structure. In the EPDM-based insulators without silica, a loose char layer can be formed during the ablation, which is formed based on the carbonized AF skeleton (Figure 2e and Figure 7a). The introduction of MWCNTs can form the micro networks to fix many micro char residues around the AF skeleton (Figure 2f and Figure 7b), which can improve the compactness of char layers, and more micro networks are formed with the increase in MWCNTs content, which results from the evolution of char layer morphology from loose structure to porous structure (Figure 2g and Figure 7c). The denser porous structure can more efficiently block heat flow than the loose structure in char layers. Meanwhile, the introduction of MWCNTs can efficiently enhance the oxidation resistance and strength of char layers by accelerating the graphitization of pyrolysis carbon (Figure 5 and Figure 6). Thus, the ablative resistance of EPDM-based insulators gradually improves as the MWCNTs increase to certain content. However, the increase in the heat transfer rate and the reduction of pore diameter in char layers occur simultaneously, which can make char layers subject to more drastic internal erosion [10,11], resulting in the decrease in ablative resistance in EPDM-based insulators with high MWCNTs content.

In the EPDM-based insulators with silica, silica can form liquid silica film by melting to increase the overflow resistance of pyrolysis gas and promote the deposition of pyrolysis gas [8,18]. Thus, a dense char layer based on the carbonized EPDM and phenolic resin can be formed by the deposition of pyrolysis gas (Figure 3e and Figure 7d). However, the addition of MWCNTs with low content can accelerate the pyrolysis of the insulators due to their high thermal conductivity, resulting in the formation of massive pores on the char layers’ surface (Figure 3f and Figure 7e). Additionally, the porous structure intensifies the intrusion of high-speed heat flow and the thermal oxidation erosion of the char layers, which leads to the decrease in ablative resistance of the insulators, and the big pores in the char layers gradually become smaller with the increase in MWCNTs content (Figure 3g and Figure 7f), since MWCNTs can more efficiently accelerate the carbothermal reduction reaction of silica to increase char residue. Furthermore, the introduction of MWCNTs with high content can form silicon carbide to enhance the strength of char layers and improve the oxidation resistance of char layers. Therefore, the ablative resistance of the EPDM-based insulators with silica enhances as the MWCNTs content increase.

To further verify the above ablation mechanism, alternating multilayer distribution of MWCNTs and silica in the insulators was designed by using S-C-0 and NS-C-3 (Figure 8d), and the variation of ablation resistance is the same as the previous result (MWCNTs have high ablation resistance efficiency in the insulators without silica). As shown in Figure 8a, it can be found that the multilayer insulators obviously show a lower linear ablation rate, and the linear ablation rate of 20 layers insulators decreased by 21.2% more than the Mixture insulators. On the one hand, the multilayer insulators have lower thermal conductivity than Mixture insulators (Figure 8b), which can be ascribed that the alternating multilayer structure disrupts the heat transfer network of the MWCNTs in the vertical direction. On the other hand, the multilayer insulators also have excellent mechanical properties as the local aggregation of MWCNTs and silica. In summary, the insulators with alternating multilayer structure not only have excellent thermal insulation performance and mechanical properties but can also more fully exploit the ablation resistance potential of MWCNTs.

## 4. Conclusions

In this work, the effects of MWCNTs content on the char layer structure and physicochemical reaction were investigated in EPDM-based insulators with and without silica. The introduction of MWCNTs showed efficient performance in improving the ablation resistance of EPDM-based insulators, but it also led to a rapid decrease in their thermal insulation properties. It was found that the MWCNTs with the same content have different effects on ablative resistance in EPDM-based insulators with and without silica, which can be ascribed to the effects of MWCNTs on the char layer structure. Although the char layer morphology of the two insulators gradually evolved into a dense porous structure as MWCNTs content increased, their formation mechanisms are different. The introduction of MWCNTs can promote the graphitization of carbonized EPDM matrix under the charring component without silica. When the charring components contain silica, the effects of MWCNTs tend to facilitate the carbothermal reduction reaction of silica. Moreover, with the addition of MWCNTs, many micro networks were found in char layers, which were considered the key to densifying the char layers of EPDM-based insulators. This work proposed the ablation mechanism based on the relationship between the char layers enhanced by MWCNTs and the ablative resistance of EPDM-based insulators and further exploited the ablation resistance potential of MWCNTs by designing the alternating multilayer distribution of MWCNTs and silica in the insulators, which can provide guidance for the construction of desirable char layer structure and promote the advancement of ablative composites.

## Figures and Tables

**Figure 1 polymers-14-03016-f001:**
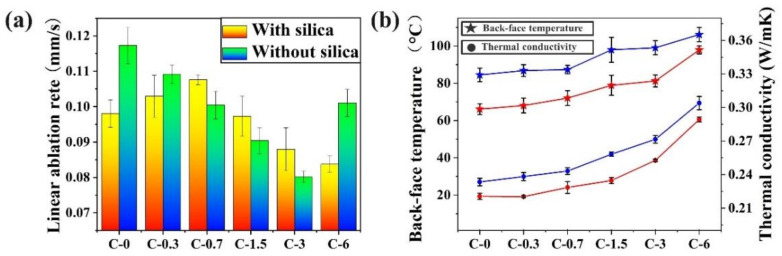
Effects of MWCNTs content on linear ablation rate (**a**) and thermal insulation performance (**b**) of the EPDM-based insulators with (Blue line) and without silica (Red line).

**Figure 2 polymers-14-03016-f002:**
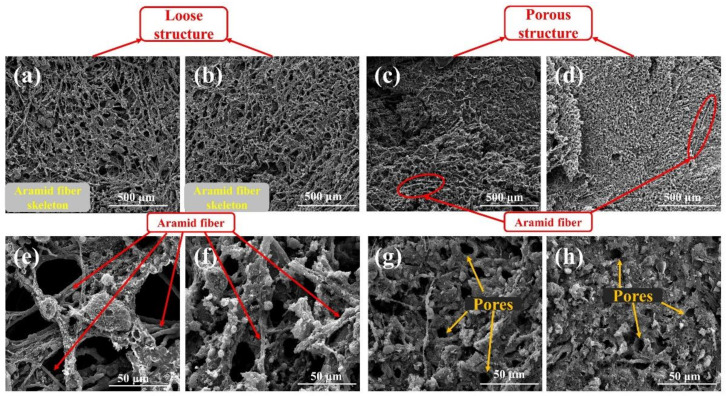
Effects of MWCNTs content on char layer surface of EPDM-based insulators without silica: NS-C-0 (400× (**a**), 3000× (**e**)), NS-C-0.3 (400× (**b**), 3000× (**f**)), NS-C-3 (400× (**c**), 3000× (**g**)) and NS-C-6 (400× (**d**), 3000× (**h**)).

**Figure 3 polymers-14-03016-f003:**
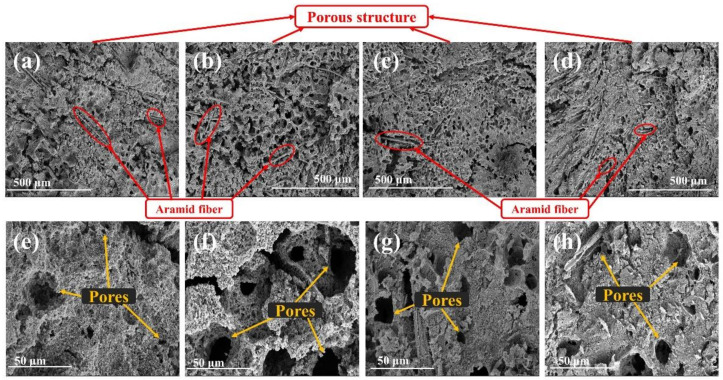
Effects of MWCNTs content on the char layer surface of EPDM-based insulators with silica: S-C-0 (400× (**a**), 3000× (**e**)), S-C-0.3 (400× (**b**), 3000× (**f**)), S-C-3 (400× (**c**), 3000× (**g**)), S-C-6 (400× (**d**), 3000× (**h**)).

**Figure 4 polymers-14-03016-f004:**
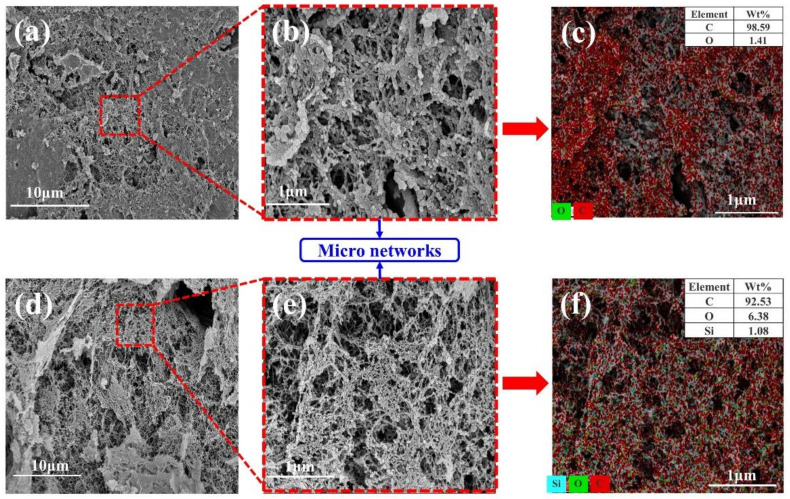
Analysis of micro networks in the char layer: NS-C-3 (**a**–**c**) and S-C-3 (**d**–**f**).

**Figure 5 polymers-14-03016-f005:**
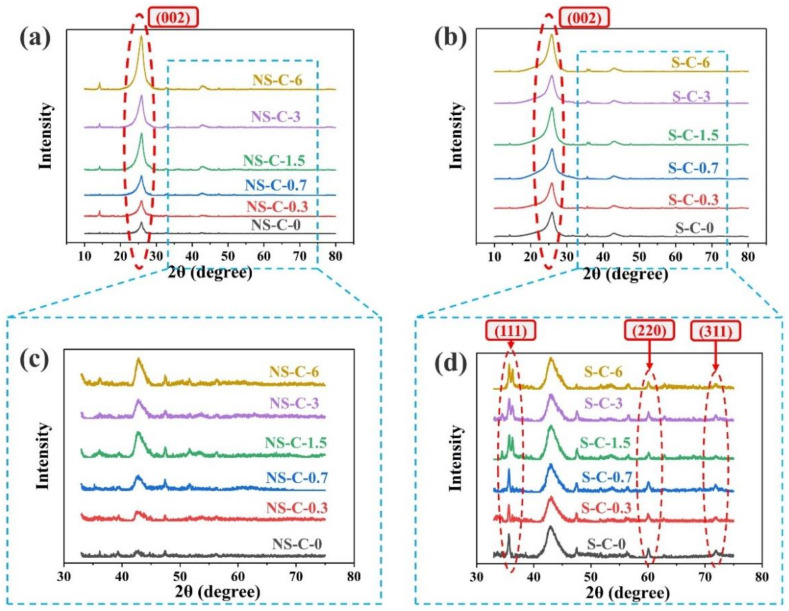
Effects of MWCNTs content on the X-ray diffraction profiles of char layers: group NS (**a**,**c**) and group S (**b**,**d**).

**Figure 6 polymers-14-03016-f006:**
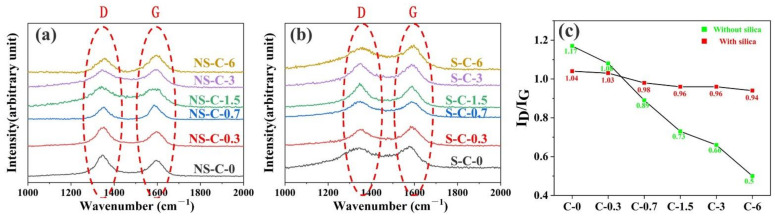
Effects of MWCNTs content on the Raman spectrum of char layers: group NS (**a**), group S (**b**), and I_D_/I_G_ (**c**).

**Figure 7 polymers-14-03016-f007:**
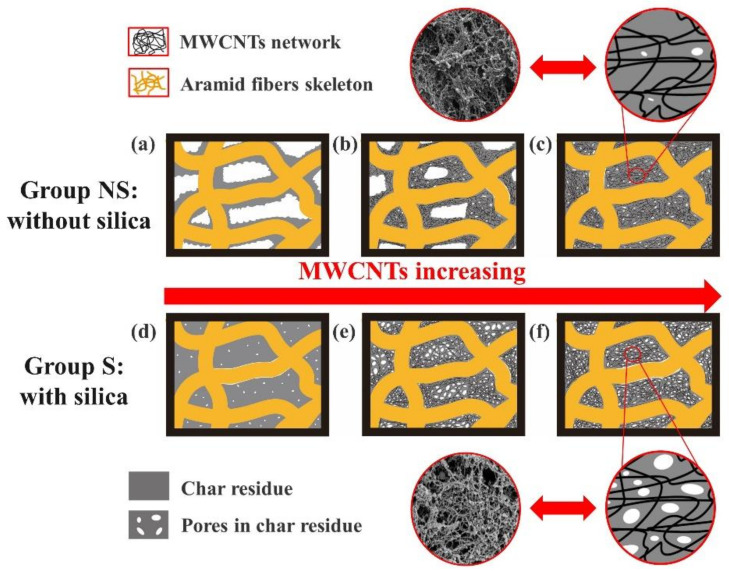
Effects of MWCNTs on char layer structure in EPDM-based insulators with ((**a**) without MWCNTs, (**b**) with low content MWCNTs, (**c**)) and without silica ((**d**) without MWCNTs, (**e**) with low content MWCNTs, (**f**)).

**Figure 8 polymers-14-03016-f008:**
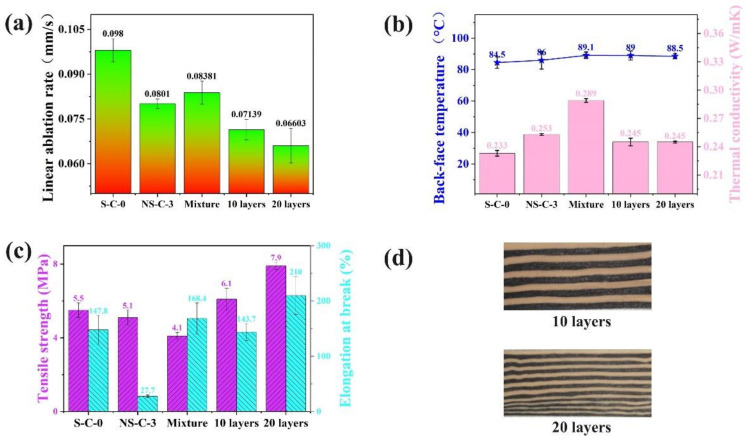
The ablation resistance (**a**), thermal insulation performance (**b**), mechanical properties (**c**) and morphology (**d**) of multilayer insulators.

**Table 1 polymers-14-03016-t001:** Effects of MWCNTs content on thickness and compress properties of char layer.

Samples	Char Layer Thickness (mm)	Compress Stress (MPa) Compress Strain 10%
S-C-0	1.57 ± 0.06	0.50 ± 0.15
S-C-0.3	1.62 ± 0.04	0.47 ± 0.22
S-C-0.7	1.76 ± 0.06	1.26 ± 0.22
S-C-1.5	1.90 ± 0.03	1.65 ± 0.24
S-C-3	2.02 ± 0.02	2.07 ± 0.04
S-C-6	2.03 ± 0.02	2.11 ± 0.08
NS-C-0	1.60 ± 0.11	0.44 ± 0.13
NS-C-0.3	1.89 ± 0.07	0.69 ± 0.23
NS-C-0.7	1.98 ± 0.04	0.93 ± 0.12
NS-C-1.5	1.92 ± 0.05	1.12 ± 0.11
NS-C-3	1.93 ± 0.07	1.16 ± 0.27
NS-C-6	1.84 ± 0.14	1.15 ± 0.25

## Data Availability

The data presented in this study are available on request from the corresponding author.

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
