# Peer review of "Effects of MWCNTs on Char Layer Structure and Physicochemical Reaction in Ethylene Propylene Diene Monomer Insulators"

_polymers, 2022, doi:10.3390/polym14153016_

Round 1
Reviewer 1 Report
The manuscript titled “Effects of MWCNTs on char layer structure and physicochemical reaction in ethylene propylene diene monomer insulators”, reports the investigation of the effects of MWCNTs on char layer structure and ablative resistance in EPDM-based insulators with and without silica. The present manuscript shows some novelties and I think that the manuscript is suitable for publication in Polymers, after a careful and major revisions. Detailed comments are present in the attached PDF file.

Reviewer 2 Report
In this paper, the authors study the effects of MWCNTs on char layer structure and ablative resistance in different EPDM-based insulators with and without silica. They show that adding MWCNTs can substantially reduce the linear ablation rate of EPDM-based insulators without silica, while higher content of MWCNTs is required to obtain similar results in EPDM-based insulators with silica. They also show that the char layer morphology of the two insulators gradually evolved into a dense porous structure as MWCNTs content increased.
The effect of MWCNT fillers on insulators has been widely investigated. In this regard, the paper can have limited novelty. However, it adds useful data to the literature and reports a well-conducted work, that is worthy of publication.
Here are some points that deserve further consideration:
- In the introduction, the authors should better motivate and highlight the novelty of their work. Why focus on composites with MWCNTs and Silica as they are already been treated in the literature?
- EPDM composites are considered the ideal insulators in combustion chamber of the solid rocket motor. Are there any other applications of these composites?
- “Fumed silica (Average particle size, 5 μm) was supplied by Shijiazhuang Ruituo Chemical Technology Co., Ltd.” Is it 5 um? Do the authors have any further information about the silica particle size?
- “In the EPDM-based insulators with silica, silica can form liquid silica film by melting 319 to increase the overflow resistance of pyrolysis gas and promote the deposition of pyrolysis gas [8, 18].” Which temperature is needed for that?
Round 2
Reviewer 1 Report
The authors responded well to all my suggestions for improving the manuscript. It can now be accepted for publication in this form.